# Evaluation of conditional treatment effect of salt stress on tomato sugar content using causal machine learning: A pilot study

Isao Goto *, Shizuka Abiko, Shiori Sugiura, Ai Furudate, Airi Suzuki, Aki Hayashi, Daiki Suzuki, Kaori Kikuchi*

School of Food Industrial Sciences, Miyagi University, Sendai, Japan

* gotoui@myu.ac.jp (IG); kkikuchi@myu.ac.jp (KK)

## Abstract

Exposing tomatoes to salt stress has been reported to increase the fruit sugar content (°Brix); however, the causal impact of this treatment under varying environmental conditions remains unclear. In this pilot study, a causal inference analysis was conducted using a Causal Tree to analyze the factors that influence the effect of salt-stress treatment on improving the °Brix in tomatoes. Data were collected from a single greenhouse using one cultivar over multiple cultivation periods, totaling 707 fruits. Propensity score matching was applied to reduce the covariate imbalance between the salt (NaCl) treatment and control groups. Using a Causal Tree, conditional average treatment effects were then estimated to assess heterogeneity in treatment impact based on environmental variables, such as temperature, vapor pressure deficit (VPD), and photosynthetically active radiation (PAR). Treatment with NaCl significantly increased °Brix compared to the control, but the magnitude of the effect varied depending on the environmental conditions. The Causal Tree analysis based on the accumulated cultivation data identified specific combinations of environmental factors under which the impact of NaCl treatment on °Brix enhancement was more pronounced. We estimated the Conditional Average Treatment Effect (CATE), defined as the difference in the proportion of producing high-sugar fruits (°Brix > 6%), between the NaCl treatment and control groups. For instance, the estimated CATE reached as high as 0.75 when the temperature was less than 25°C, the VPD was at least 0.84 kPa, and the PAR was below 13 mol·m⁻²·d⁻¹. In contrast, the CATE dropped to 0.031 when the temperature was between 20 and 25°C and the VPD was below 0.84 kPa. These results suggest that the interplay between temperature and VPD significantly modulates the efficacy of NaCl treatment for increasing tomato sugar content.

## Introduction

Tomatoes are one of the most important crops in the world, with a global production of over 180 million tons in 2021 [1,2]. They are a valuable source of vitamins,

**Data availability statement:** Response to Data Availability Inquiry We confirm that all raw data required to replicate the results of this study have been deposited in a public repository and are openly available in Figshare at https://doi.org/10.6084/m9.figshare.30844622. The deposited dataset includes the complete set of values underlying all statistical analyses, figures, and tables presented in the manuscript. Therefore, it constitutes the full minimal data set as defined by PLOS ONE.

**Funding:** The author(s) received no specific funding for this work.

**Competing interests:** The authors have declared that no competing interests exist.

minerals, and antioxidants and are consumed in various forms, including fresh, processed, and canned [3]. Sugar content is considered one of the most important factors affecting tomato fruit quality and consumer satisfaction [4]. Tomatoes with a high sugar content are popular in Japan, China, and Korea, and in Japan in particular, they are considered luxury items and are often traded at high prices. Therefore, a cultivation method that applies water stress to concentrate components within the fruit is used to increase the sugar content in tomatoes [5,6].

In hydroponic cultivation, sodium chloride is added to the nutrient solution to increase the osmotic pressure in the root zone, thereby restricting water absorption [7,8]. Applying salt stress, rather than limiting irrigation, is considered a more efficient cultivation method because it provides uniform water stress across the entire root zone and allows for more precise adjustment of stress levels. However, even if constant salt stress is applied during cultivation, the fruit sugar content is not constant. Even when cultivated without stress, the sugar content of tomatoes fluctuates widely throughout the year [9], largely due to the strong influence of environmental factors on both sugar accumulation and stress responses. Factors that determine the sugar content of fruits include photosynthesis, which synthesizes sugar; respiration, which consumes sugar; and the amount of water supply, which affects sugar concentration. Environmental conditions, such as temperature, solar radiation, and humidity, and the duration of exposure to them, significantly influence these factors. In particular, temperature and solar radiation are important factors that determine the balance between photosynthesis and respiration, and humidity affects water absorption and sugar concentration [10,11]. Moreover, these factors do not act independently but interact, making the relationship complex [12]. Therefore, to stabilize sugar content, it is necessary to analyze in detail how environmental factors such as temperature, solar radiation, and humidity interact with photosynthesis, respiration, and water uptake throughout the seasons. It is also important to understand how these interactions vary across different growth stages and durations, leading to fluctuations in sugar content, to establish management methods based on this analysis.

In this study, tomatoes were cultivated year-round and causal inference was used to examine the treatment effect of salinity stress on sugar content. Specifically, tomatoes were cultivated in nutrient solutions at different times after adjusting for the sowing time. Two groups were established for each sowing period: a salt-stress treatment group and a control group (without salt stress). For tomatoes matured under the same cultivation conditions (environment), the percentage of tomatoes with a sugar content above 6 °Brix (or °Bx, i.e., the percent by weight of sugar solids in a pure sucrose solution) was calculated for both treatment groups. Factors influencing the percentage difference between the two groups were analyzed using causal inference. Based on these results, we aimed to identify environmental factors and tree conditions that enhance the effects of salt stress.

## Materials and methods

### Plant materials and growing conditions

Tomato (*Solanum lycopersicum* L. "Momotaro York"; Takii & Co., Ltd., Japan) seeds were sown nine times between March 2016 and June 2022 (Table 1). The seeds were

**Table 1. Sowing dates and average anthesis and cultivation end dates (n = 12) for each cultivation period.**

| Cultivation period | Sowing date (year) | Anthesis[y] | | | End-of-cultivation date[z] |
|---|---|---|---|---|---|
| | | Truss1 | Truss2 | Truss3 | |
| A | 1/16 (2017) | 3/30 | 4/6 | 4/13 | 7/4 |
| B | 2/22 (2022) | 5/1 | 5/10 | 5/16 | 7/19 |
| C | 3/16 (2016) | 5/24 | 5/28 | 6/6 | 8/8 |
| D | 4/22 (2021) | 6/25 | 7/4 | 7/12 | 8/30 |
| E | 5/16 (2016) | 7/11 | 7/16 | 7/16 | 9/30 |
| F | 6/17 (2022) | 8/12 | 8/22 | 8/25 | 11/11 |
| G | 7/16 (2016) | 9/6 | 9/14 | 9/14 | 12/28 |
| H | 9/16 (2017) | 11/21 | 12/6 | 12/6 | 5/1 |
| I | 11/16 (2017) | 2/20 | 2/23 | 3/8 | 6/2 |

[y]The date of 2-methyl-4-chlorophenoxyacetic acid application was regarded as the date of anthesis.

[z]The cultivation end date was defined as the day when all fruits were harvested.

sown in rockwool plugs (Kiemplug; Grodan Delta, The Netherlands) and allowed to germinate in a temperature-controlled growth chamber at 25/20°C (day/night). After germination, seedlings were transplanted into rockwool cubes (75×75×65 mm; Grodan Delta) and allowed to grow in a greenhouse supplied with a commercial nutrient solution, Otsuka A (OAT Agrio Co., Ltd., Japan), adjusted to an electrical conductivity (EC) of 0.8 dS/m and pH of 6.0–6.5. Just before first-truss flowering, young plants were transplanted into a nutrient film technique system supplied with the commercial nutrient solution Otsuka SA, adjusted to an EC of 1.2 dS/m. Once the first flower on each truss had fully opened, all flowers on the truss were sprayed with a commercial 4-CPA (2-methyl-4-chlorophenoxyacetic acid) formulation (Tomato-Ton; Green Japan Co., Ltd., Japan). The solution was diluted to concentrations of approximately 30 mg L$^{-1}$ at low temperatures (<20°C) and 15 mg L$^{-1}$ at high temperatures (≥20°C), following the manufacturer's instructions. No flower thinning was performed, and the number of flowers per truss developed naturally. All lateral shoots were removed as they appeared, and the plants were trained to a single stem and pinched above the third truss with two true leaves over the truss. The plants were spaced at 30 cm within rows, resulting in a planting density of 2,380 plants per 10 a. The cultivation was conducted under sparse planting conditions, and the leaf area index (LAI) during the fruit development period was approximately 3.5. Cultivation was terminated after the fruits were harvested from the third flower cluster. Cultivation periods A to I are defined in Table 1, based on the sowing dates. The dates when flowers of the first, second, and third trusses began anthesis and the cultivation termination dates are shown for each cultivation period. The dates of anthesis for each truss represent the average flowering dates of the 12 plants surveyed in each cultivation period. Throughout all cultivation periods, the greenhouse was automatically ventilated when the air temperature exceeded 25°C, and heating was activated to maintain temperatures above 10°C in winter. As shown in Table 3, the mean daily air temperature, average vapor pressure deficit (VPD), and photosynthetically active radiation (PAR) from flowering to harvest clearly differed among cultivation periods. The temperature, relative humidity, and illuminance were measured every 10 min using a data logger (RS-13L; ESPSC MIC Corp., Ltd., Japan). Temperature and humidity sensors were installed inside a ventilated PVC tube (φ5 cm, double-tube type) equipped with a small DC fan to ensure forced airflow, following the design principle described previously [13]. This setup was placed horizontally within the plant canopy to measure air temperature and humidity under well-ventilated conditions while minimizing the influence of solar radiation. The illuminance sensor was installed at the top of the cultivation shelf (height 1.8 m). The average temperature and relative humidity were calculated for each fruit over the period from anthesis to harvest. The illuminance measurements taken every 10 min were converted to a per-second basis, and the integral illuminance was calculated for the period from anthesis to harvest. VPD and daily integrated PAR were calculated from the average air temperature, relative humidity, and illuminance measured between 6:00 and 18:00. VPD was calculated as the difference between the saturation vapor pressure and the actual vapor pressure:

$$e_s = 0.6108 \times \exp\left(\frac{17.27 \times T}{T + 237.3}\right) \tag{1}$$

$$e_a = e_s \times \frac{RH}{100} \tag{2}$$

$$VPD = e_s - e_a \tag{3}$$

PAR was estimated from illuminance (lux) using photosynthetic photon flux density (PPFD) and empirical relationship for daylight:

$$PPFD = \frac{lux}{54} \tag{4}$$

$$PAR = PPFD \times 3600 \times 24 \div 10^6 \tag{5}$$

The environmental data (temperature, VPD, and PAR) for each fruit were used as covariates (X) in the causal inference analysis as shown in Table 2.

## Permits and approvals

All experiments were conducted in the experimental greenhouse of Miyagi University. Access to the field site was authorized by the Faculty of Food Industry, Miyagi University, which manages and operates the greenhouse facilities. No external permits were required because the study was performed entirely within a university-operated experimental facility and did not involve any endangered or protected species. The tomato plants used in this study were a commercial cultivar ('Momotaro York', Takii & Co., Ltd., Japan).

**Table 2. Dataset characteristics before and after propensity score matching.**

| | Variable | | | Dataset A | | Dataset B | |
|---|---|---|---|---|---|---|---|
| | | | | Control | Treatment | Control | Treatment |
| n | | | Fruits | 323 | 349 | 301 | 301 |
| Cultivation period | A | | Fruits | 19 | 30 | 19 | 19 |
| | B | | Fruits | 42 | 42 | 42 | 30 |
| | C | | Fruits | 56 | 75 | 56 | 68 |
| | D | | Fruits | 15 | 18 | 12 | 13 |
| | E | | Fruits | 43 | 48 | 43 | 43 |
| | F | | Fruits | 30 | 36 | 29 | 34 |
| | G | | Fruits | 55 | 52 | 50 | 48 |
| | H | | Fruits | 36 | 30 | 26 | 28 |
| | I | | Fruits | 27 | 18 | 24 | 18 |
| Truss | F1 | | Fruits | 111 | 122 | 107 | 102 |
| | F2 | | Fruits | 104 | 112 | 89 | 97 |
| | F3 | | Fruits | 108 | 115 | 105 | 102 |
| °Brix | | mean (SD) | % | 5.48 (0.68) | 6.36 (0.91) | 5.49 (0.67) | 6.38 (0.92) |
| Temperature | Tem. | mean (SD) | °C | 20.7 (3.4) | 21.1 (3.2) | 20.9 (3.3) | 21.0 (3.3) |
| Vapor Pressure Deficit | VPD | mean (SD) | kPa | 0.92 (0.15) | 0.93 (0.15) | 0.93 (0.15) | 0.93 (0.15) |
| Photosynthetically Active Radiation | PAR | mean (SD) | mol·m⁻²·d⁻¹ | 11.1 (2.9) | 11.3 (2.9) | 11.1 (2.9) | 11.1 (2.9) |

## Salinity treatments

In the control treatment, the EC level was maintained at 1.2 dS/m from transplantation until the end of cultivation. In the NaCl treatment, NaCl was added to the standard nutrient solution at first-truss flowering and gradually increased to the desired EC of 4.0 dS/m (corresponding to 25 mM NaCl) at second-truss flowering. The EC was maintained at 4.0 dS/m until all the fruits from the third truss were harvested.

Previous studies have evaluated the effect of salt concentration on tomato sugar content, with reports indicating that higher salt stress leads to increased sugar content [14–17]. Consequently, many studies have used a salt concentration of 50 mM NaCl (≈ 8 dS/m) to reliably increase the sugar content [18–20]. However, growth and yield significantly decrease when the salt concentration exceeds a certain level [21]. Various strategies have been employed to address this trade-off, such as limiting the timing of salt stress or gradually increasing the salt stress [22].

We aimed not to evaluate the effect of the salt concentration but to elucidate how the cultivation environment influences the effect of salt stress on increasing sugar content. The stress associated with high salt concentrations may be less affected by cultivation conditions; therefore, in this study, we used a lower salt concentration (EC: 4 dS/m, corresponding to 25 mM NaCl) than typically used in previous studies. We expect the impact of environmental conditions to be more pronounced at this level of salt stress.

## Fruit yield and quality analyses

Marketable fruits with fresh weights of 50 g or more from the surveyed plants' first, second, and third trusses were selected and harvested when they reached the red stage. After harvesting, the fruits were weighed to determine their fresh weights. The fruits were then split in half. One half was used to measure total soluble solids (°Brix) content using a hand refractometer (PAL-BX|ACOD3; Atago Co., Ltd., Tokyo, Japan) and the other half was oven-dried at 80°C for 48 h and the dry weight was measured. The total number of sampled fruits was 707, and the °Brix values were used for subsequent analyses.

## Analysis

**Data selection and analysis flow.** Dataset A was created by applying basic data cleaning procedures, such as removing missing values, to all fruit (n = 707) and environmental data collected (Fig 1). We examined the data characteristics of Dataset A, including the descriptive statistics of each variable. Propensity score matching was performed to adjust for potential biases between the control and treatment groups, resulting in Dataset B (n = 602). Using Dataset B, the conditional average treatment effects (CATEs) on salt stress were estimated using a Causal Tree.

**Propensity score matching.** We performed 1-to-1 propensity score matching without replacement to pair fruits in the control and treatment groups. The covariates used for estimating the propensity score were all variables listed in Table 2 except for the outcome variable (°Brix). Specifically, the variables included Truss, mean temperature, VPD, and PAR. Propensity scores for treatment were estimated using a logistic regression model. Matching was performed using a caliper width of 0.1 standard deviations of the logit of the propensity score. A successful balance between the groups was confirmed by an absolute standardized mean difference of < 0.1.

**Causal tree.** We applied a Causal Tree in this study to estimate the CATE, allowing for heterogeneous NaCl treatment effects [23]. A Causal Tree is a decision tree-based approach that provides interpretable subgroup-level treatment effects and captures nonlinear treatment heterogeneity.

**Data and variables.**

- Outcome variable ($Y$): °Brix (1 = above 6%, 0 = below 6%)

In the absence of a widely accepted threshold for this evaluation metric, a provisional cutoff (6%) was determined based on the observed distribution in the treatment and control groups.

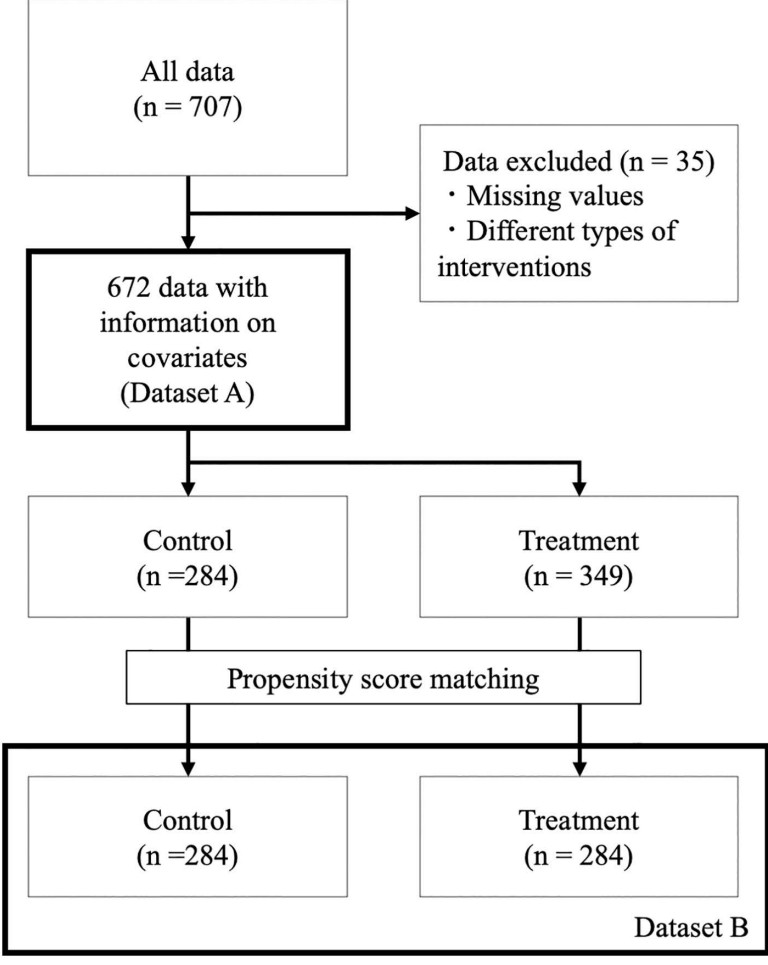

**Fig 1. Flow of study data selection.** The 672 fruits after data cleaning were prepared as Dataset A and the 602 fruits after propensity score matching as Dataset **B.**

- Treatment variable ($W$): NaCl treatment (1 = treated, 0 = control)

- Covariates ($X$): temperature (Tem.), VPD, PAR, number of trusses (Truss; the first, second, and third trusses were designated F1, F2, and F3, respectively)

   **Model specification.** The Causal Tree method was implemented using the causalTree package in R (https://github.com/susanathey/causalTree). The hyperparameters were set as follows:

- Splitting rule: Causal Tree (split.Rule = "CT")

- Cross-validation method: Causal Tree (cv.option = "CT")

- Minimum leaf size (subgroup size): 30 (minsize = 30)

- Honest splitting: disabled (split honest = FALSE)

The choice of a minimum subgroup size of 30 was based on prior literature, which ensured that each subgroup contained enough observations for stable effect estimation [23]. Honest splitting, a technique that separates data into training and validation subsets, was not performed to preserve the entire dataset for estimation purposes.

**Estimation of CATE.** The Causal Tree method partitions the data based on covariates $X$ and estimates the CATE within each subgroup node using the following equation:

$$CATE(X) = E[Y(1)|X] - E[Y(0)|X], \tag{6}$$

where $E[Y(1)|X]$ represents the expected outcome for the treated group conditioned on $X$ and $E[Y(0)|X]$ represents the expected outcome for the control group conditioned on $X$. The outcome variable $Y$ was a binary indicator representing whether the fruit had high sweetness (1 if °Brix > 6%, otherwise 0). Accordingly, the CATE represents the difference in the proportion of producing high-sugar fruits (°Brix > 6%) between the NaCl treatment and control groups, conditional on the covariates $X$. This approach enables the analysis of how the impact of salt-stress treatment varies across different subgroups, considering factors such as environmental conditions.

**Statistical analysis.** To compare the means between the two groups, Welch's t-test was employed because one group did not meet the assumption of normality. The assumption of normality was assessed using the Shapiro–Wilk test. In addition to the p-values, mean differences with 95% confidence intervals were reported to indicate the magnitude and uncertainty of the effect.

**Analytical environment.** All analyses were performed using R version 4.2.2 (R Foundation for Statistical Computing, Vienna, Austria). The following R packages were used: effectsize (1.0.0), devtools (2.4.5), tidyverse (1.3.1), Matching (4.10.8), and ggplot2 (3.4.0). We used the causalTree package (version 1.0) developed by Athey and Imbens [23] for model construction. This package was installed in the susanathey/causalTree GitHub repository.

## Results

The sample size for Dataset A was 672, with the control and treatment groups consisting of 323 and 349 samples, respectively (Fig 1). Sample sizes varied across cultivation periods (Table 2). After adjusting for environmental covariates, the transplant-to-harvest period differed by an average of 2.5 days among periods (data not shown). Propensity score matching was applied to ensure a more balanced comparison, resulting in reduced variation in most covariates between the two Dataset B groups (Table 2). NaCl treatment significantly increased the °Brix compared with the control (p < 0.01). The mean °Brix of the control was 5.49 (95% CI: 5.41–5.56) and that of salt-stress treatment was 6.38 (95% CI: 6.27–6.48) (Fig 2). The mean difference was 0.89 (95% CI: 0.76–1.01), suggesting a higher °Brix under salt stress.

We examined whether the effectiveness of the NaCl treatment varied depending on the cultivation period. Fig 3 illustrates the relationship between the salt-stress treatment and °Brix values across different cultivation periods. NaCl treatment consistently increased the fruit °Brix in all cultivation periods; however, the magnitude of its effect varied with the cultivation period. Cultivation periods A, B, C, E, G, H, and I showed meaningful effects, whereas D and F yielded only negligible differences (S1 Table). Table 3 summarizes the mean and standard deviation of the room temperature, VPD, and PAR in the control and treatment groups during each growing period. There were only minor differences in environmental conditions between the control and treatment plots within each period. In contrast, substantial differences in environmental conditions were observed across cultivation periods. For instance, while the average difference in temperature between the control and treatment was less than 0.5°C in cultivation periods B and C, the difference in VPD exceeded 0.12 kPa.

We also examined how Causal Tree-derived subgroups corresponded to the cultivation period. While D was assigned to a single terminal node because of its limited sample size, the other periods were represented across multiple subgroups. Furthermore, each Causal Tree subgroup included samples from multiple cultivation periods (Table 4), supporting

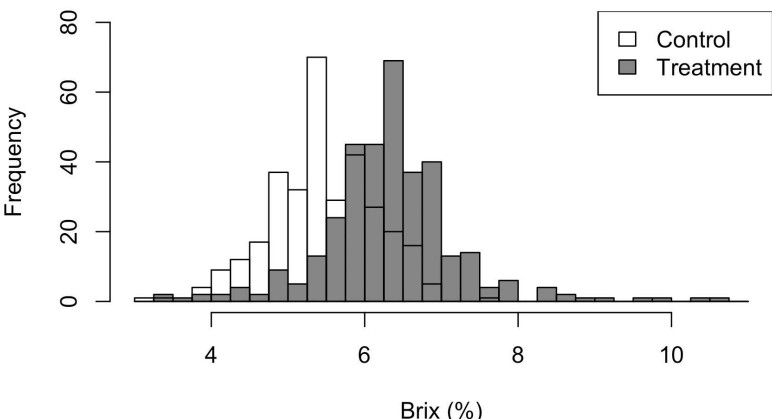

**Fig 2. Overlapping histograms showing the distribution of tomato °Brix in the control (white) and salt-stressed treatment (gray) groups.** The x-axis represents °Brix values, and the y-axis indicates the number of fruits (frequency).

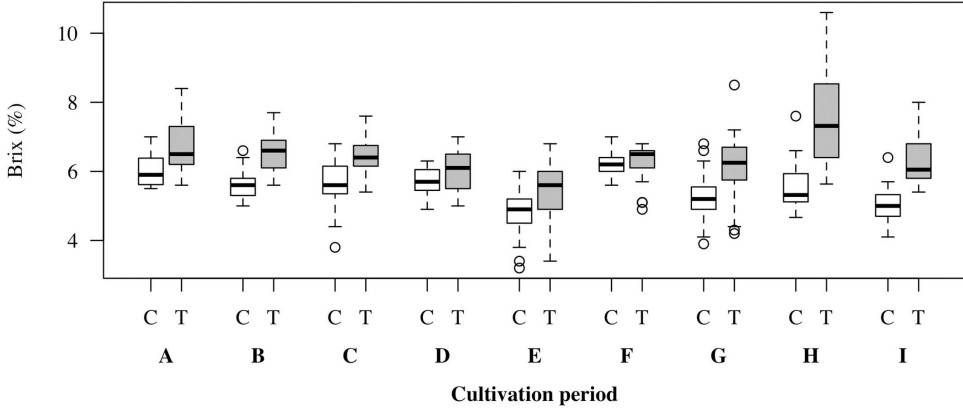

**Fig 3. Box plots of tomato °Brix for the control (white) and salt-stressed treatment (gray) groups across nine cultivation period (A–I).** The boxes represent the interquartile range (IQR) from the first (Q1) to the third (Q3) quartile, with the central line indicating the median. Whiskers extend to data points within 1.5×IQR, and outliers are plotted as circles (○).

**Table 3. Mean (SD) of temperature, vapor pressure deficit (VPD), and photosynthetically active radiation (PAR) in the greenhouse for the control and NaCl treatment groups across each cultivation period, and differences between the group means.**

| Cultivation period | Temperature (°C) | | | Vapor Pressure Deficit (kPa) | | | Photosynthetically active radiation (mol·m⁻²·d⁻¹) | | |
|---|---|---|---|---|---|---|---|---|---|
| | Control | Treatment | Difference | Control | Treatment | Difference | Control | Treatment | Difference |
| A | 18.7 (0.3) | 18.7 (0.3) | 0.0 | 1.16 (0.02) | 1.17 (0.02) | 0.01 | 16.4 (0.2) | 16.4 (0.2) | 0 |
| B | 22.3 (0.7) | 22.4 (0.6) | 0.1 | 1.05 (0.04) | 1.07 (0.07) | 0.02 | 12.3 (0.5) | 12.4 (0.7) | 0.1 |
| C | 22.7 (0.2) | 22.7 (0.3) | 0.0 | 0.93 (0.06) | 0.95 (0.1) | 0.02 | 13.1 (0.6) | 13.3 (1) | 0.2 |
| D | 25.4 (0.2) | 25.1 (0.2) | 0.3 | 0.87 (0.06) | 0.85 (0.06) | 0.02 | 13.2 (0.9) | 13.0 (0.8) | 0.2 |
| E | 25.1 (0.2) | 25.2 (0.2) | 0.1 | 1.05 (0.06) | 1.05 (0.04) | 0 | 8.5 (1) | 8.4 (0.7) | 0.1 |
| F | 22.3 (0.99) | 22.3 (0.9) | 0.0 | 0.72 (0.03) | 0.71 (0.03) | 0.01 | 8.9 (0.7) | 8.9 (0.7) | 0 |
| G | 18.0 (1.1) | 18.2 (1.1) | 0.2 | 0.78 (0.03) | 0.78 (0.03) | 0 | 7.9 (0.4) | 8.0 (0.4) | 0.1 |
| H | 14.9 (0.2) | 14.9 (0.3) | 0.0 | 0.78 (0.04) | 0.79 (0.05) | 0.01 | 9.3 (1) | 9.3 (1.1) | 0 |
| I | 16.9 (0.4) | 16.9 (0.3) | 0.0 | 1.07 (0.02) | 1.08 (0.03) | 0.01 | 15.4 (0.4) | 15.4 (0.6) | 0 |

**Table 4. Number of fruits observed during the growing season for each subgroup node in the causal tree.**

| Cultivation period | Subgroup | | | | | | |
|---|---|---|---|---|---|---|---|
| | 1 | 2 | 3 | 4 | 5 | 6 | 7 |
| A | 0 | 0 | 0 | 0 | 38 | 0 | 0 |
| B | 0 | 0 | 0 | 0 | 0 | 4 | 68 |
| C | 0 | 18 | 0 | 0 | 0 | 86 | 20 |
| D | 21 | 1 | 0 | 0 | 0 | 3 | 0 |
| E | 69 | 0 | 0 | 0 | 0 | 0 | 17 |
| F | 0 | 63 | 0 | 0 | 0 | 0 | 0 |
| G | 0 | 4 | 60 | 32 | 0 | 0 | 2 |
| H | 0 | 0 | 0 | 35 | 0 | 0 | 19 |
| I | 0 | 0 | 0 | 0 | 27 | 15 | 0 |

the interpretation that subgrouping was primarily driven by environmental and contextual variables rather than seasonal classification alone.

The outcome variable was provisionally dichotomized based on °Brix values, classifying fruits as either < 6% or ≥ 6% to evaluate the effect of the salt-stress treatment. First, the CATE was estimated using the Causal Tree method; its structure is illustrated in Fig 4. After propensity score matching, the overall average treatment effect was 0.49. However, the CATE differed across subgroups; for instance, it was 0.2 when the highest splitting variable, temperature, was ≥ 25°C and 0.55 when temperature was < 25°C. The highest CATE of 0.75 (95% CI: 0.63–0.86) was observed under the following conditions: Tem. < 25°C, VPD ≥ 0.84 kPa and PAR < 13 mol·m$^{-2}$·d$^{-1}$ (S2 Table). Conversely, the lowest CATE estimate of 0.031 (95% CI: −0.14–0.20) was found under temperature < 25°C, VPD < 0.84 kPa and temperature ≥ 20°C (S2 Table). Fig 5 illustrates the relationship between the salt-stress treatment and °Brix values across subgroups. In Subgroup 1, the median °Brix did not exceed 6% in either the control or treatment group. In Subgroup 2, both groups exhibited median

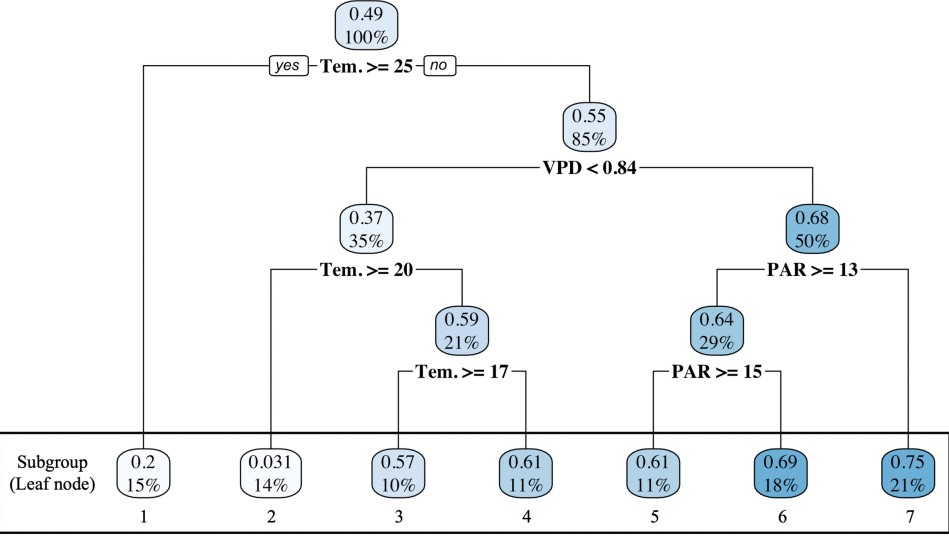

**Fig 4. Causal Tree illustrating factors influencing tomato °Brix.** Each node represents a splitting variable, with data partitioned based on the threshold values. The subgroups indicate the classified groups and their respective proportions. The values above and below each node denote the CATE and the proportion of Dataset B, respectively.

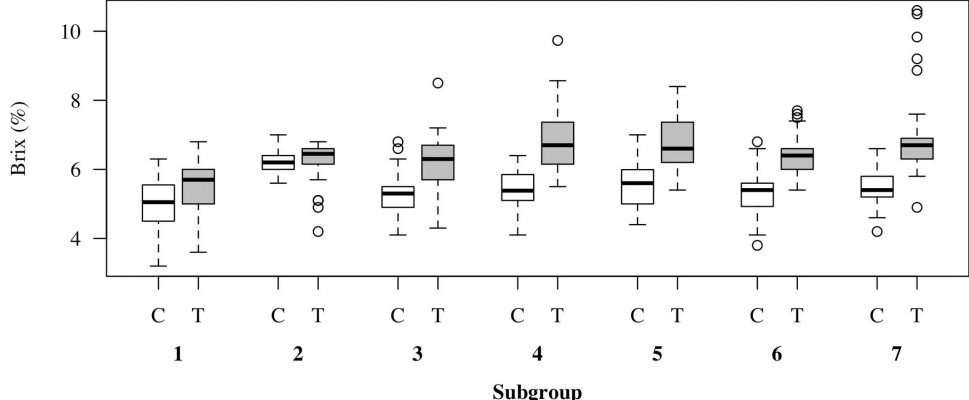

**Fig 5. Box plots of tomato °Brix for the control (white) and salt-stressed treatment (gray) groups across seven subgroups (1–7).** The boxes represent the interquartile range (IQR) from the first (Q1) to the third (Q3) quartile, with the central line indicating the median. Whiskers extend to data points within 1.5×IQR, and outliers are plotted as circles (○).

°Brix greater than 6%, and no substantial difference was observed between the treatment and control conditions. In Subgroups 3–7, the median °Brix consistently exceeded 6% in the treatment group, whereas the control group remained at approximately 5%. These findings suggest that environmental factors, such as temperature, VPD, and PAR, influence the effectiveness of NaCl treatment on °Brix. Additionally, we examined the relationship between cultivation periods and the subgroups derived from the Causal Tree. Cultivation periods A and F were classified into only one subgroup, whereas datasets from other growing periods were distributed across multiple subgroups (Table 4). Subgroup 3 was classified into only one cultivation period, while other Causal Tree subgroup included samples from multiple cultivation periods (Table 4). This indicates that the subgrouping based on the Causal Tree was not merely a result of seasonal classification but was driven by environmental and contextual differences.

## Discussion

We examined the effects of a NaCl treatment on fruit quality across different cultivation periods. First, we confirmed that NaCl treatment affected fruit quality throughout all cultivation periods. Second, we observed that the effects on fruit quality differed depending on the cultivation period. Third, we evaluated the conditional effects of treatments by incorporating environmental factors as covariates and found that temperature had the most significant impact on the treatment effects.

A causal tree is an algorithm that automatically stratifies data based on a decision tree framework within machine learning. In this study, the first split was determined by temperature, with CATE of 0.20 above 25°C and 0.55 below 25°C (a difference of 0.35). These results indicate that temperature was the primary determinant of the NaCl treatment effect, defined as the proportion of plants producing fruits with °Brix values above 6%. Under high-temperature conditions (mean daily temperature > 25°C), the soluble solid content of fruits decreased in both control and NaCl-treated plants, suggesting that elevated temperature itself served as a physiological bottleneck for sugar accumulation.

At the second node of the causal tree, the data were split at a VPD threshold of 0.84 kPa, resulting in CATE values of 0.37 and 0.68 for the respective subgroups. Thus, even under moderate temperature conditions (mean temperature < 25°C), a difference of 0.31 was observed depending on the VPD level, indicating that the strength of the transpiration driving force influenced the NaCl-induced enhancement of sugar content. In general, within the optimal temperature range for tomato (20–25°C), the balance between transpiration and photosynthesis is well maintained, whereas a VPD below approximately 0.8 kPa reduces transpiration demand and suppress the concentration effect of soluble solids [24].

The present results are consistent with this physiological understanding, suggesting that, at temperatures below 25°C, the NaCl-induced enhancement of sugar content became more pronounced when VPD exceeded 0.84 kPa.

Furthermore, under conditions of VPD < 0.84 kPa, the CATE values were 0.031 and 0.59 at temperatures above and below 20°C, respectively. As shown in Fig 5, in subgroup 2, both the NaCl-treated and control plots exhibited median °Brix values exceeding 6%, and the difference in soluble solids between the two groups was minimal, resulting in a low CATE value. Because CATE represents the difference in the proportion of fruits with °Brix ≥ 6% between the NaCl-treated and control plots, when the control group already exhibits high °Brix values, the potential for further enhancement is naturally limited, leading to lower CATE values. The temperature range of 20–25°C is considered physiologically favorable for sugar accumulation in tomato fruit due to optimal photosynthetic activity and efficient translocation of assimilates [25,26]. Moreover, conditions of VPD < 0.84 kPa (indicating moderately high humidity) likely supported stable transpiration and photosynthetic activity, thereby facilitating assimilate transport to the fruit. This interpretation is consistent with the concept that a VPD of approximately 0.7–1.0 kPa represents a "moderate transpiration load," which is regarded as optimal for greenhouse tomato cultivation [27]. Therefore, in subgroup 2, both temperature and transpiration load provided favorable conditions for sugar formation, resulting in high °Brix values even without NaCl treatment, which could explain the small CATE observed under these conditions.

In contrast, under VPD < 0.84 and Tem. < 20°C, CATE reached 0.59. In this relatively low-temperature range (Subgroups 3 and 4; Tables 3 and 4), fruit development periods tended to be longer. Compared with Subgroup 2 (20–25°C), osmotic concentration effects caused by NaCl-induced mild water restriction [15], combined with prolonged maturation and sugar accumulation time, likely enhanced the treatment effect. This finding is consistent with previous studies reporting that extended fruit development at lower temperatures contributes to higher sugar accumulation [17].

Finally, under higher VPD conditions (≥ 0.84 kPa), PAR emerged as a branching factor in the causal tree. Under such conditions, transpiration flow and photosynthesis are expected to be sufficiently activated, allowing NaCl-induced water stress and osmotic concentration effects to manifest more strongly. Therefore, the emergence of PAR as a branching factor under these conditions does not necessarily indicate a direct causal link between light intensity and sugar formation. Rather, given that the analysis was based on the threshold of 6% °Brix, PAR likely acted as a stratifying indicator reflecting how light conditions modulate the distribution boundary of fruit sugar content between the two treatments. Indeed, even under high-PAR conditions, the sugar content distributions of the control and NaCl-treated groups overlapped substantially, suggesting that light intensity was not the direct determinant of sugar-enhancement effects. Instead, PAR likely appeared as an indirect factor representing the combined influence of temperature and transpiration load (VPD). Taken together, these results suggest that the enhancement of fruit sugar content by NaCl treatment is not primarily driven by light conditions but manifests most clearly under environments where both temperature and transpiration load are balanced within physiologically optimal ranges.

This study had several limitations. First, data were collected using a single tomato cultivar from a single greenhouse site. Although the dataset was highly accurate, the findings may not be generalizable to other cultivars, greenhouse facilities, or environmental settings. Future studies should validate these results using datasets from multiple sites and cultivars under various climatic and soil conditions. Second, while the Causal Tree method offers high interpretability and facilitates the exploration of treatment effect heterogeneity, its results may be sensitive to hyperparameter selection and the limited sample size within subgroups. Third, this study was conducted as an exploratory pilot study, and the sample size used in the analysis was relatively small. Therefore, the specific patterns and associations between variables observed in the results of the Causal Tree analysis may be specific to this dataset, and their generalizability and robustness need to be verified in further large-scale studies. In particular, the stability of the Causal Tree using Honest Estimation and a detailed robustness evaluation using the bootstrap method are issues to be addressed in the future. Finally, the outcome variables were classified using a provisional °Brix threshold of 6% based on the distribution of the observed data. Depending on specific research objectives or practical quality standards in future studies, this threshold may require adjustment.

 

Furthermore, while this study focused on °Brix as an indicator of fruit quality, future research should also address the potential trade-offs between sugar enhancement and yield stability, as well as the risk of salt toxicity under varying salinity levels and across different tomato varieties.

## Conclusions

In our experimental design, mild salinity stress (25 mM NaCl) consistently increased tomato fruit sugar content (°Brix), although the magnitude of enhancement varied with environmental conditions. Causal Tree analysis revealed that temperature, VPD and PAR jointly affected the treatment effect, with the strongest enhancement occurring under a specific combination of conditions (Tem. < 25°C, VPD ≥ 0.84 kPa and PAR < 13). These findings suggest that, within the environmental range examined in this study, NaCl-induced sugar accumulation depends on the balance between temperature, light intensity and transpiration load rather than on a single factor.

Furthermore, Causal Tree analysis quantitatively illustrated how treatment efficacy changed with cultivation conditions, providing a useful framework for identifying where mild salinity may be most effective. This approach offers a data-driven basis for optimizing NaCl application strategies and could be extended to other quality indices in hydroponic tomato production.

## Supporting information

**S1 Table. Difference in average Brix of tomato fruits between control and treatment.**
(DOCX)

**S2 Table. Subgroup-Specific CATE derived from Causal Tree splits based on environmental conditions.**
(DOCX)

## Acknowledgments

We sincerely thank Dr. Yasunaga Iwasaki of Meiji University for his valuable advice on salt-stress treatment. We also thank Ms. Chisato Goto for her helpful support throughout this study.

## Author contributions

**Conceptualization:** Isao Goto, Kaori Kikuchi.

**Data curation:** Isao Goto, Shizuka Abiko, Shiori Sugiura, Ai Furudate, Airi Suzuki, Aki Hayashi, Daiki Suzuki, Kaori Kikuchi.

**Formal analysis:** Isao Goto, Kaori Kikuchi.

**Funding acquisition:** Isao Goto, Kaori Kikuchi.

**Investigation:** Isao Goto, Kaori Kikuchi.

**Methodology:** Isao Goto, Kaori Kikuchi.

**Project administration:** Isao Goto, Kaori Kikuchi.

**Resources:** Isao Goto, Kaori Kikuchi.

**Software:** Isao Goto, Kaori Kikuchi.

**Supervision:** Isao Goto, Kaori Kikuchi.

**Validation:** Isao Goto, Kaori Kikuchi.

**Visualization:** Isao Goto, Kaori Kikuchi.

**Writing – original draft:** Isao Goto, Kaori Kikuchi.

**Writing – review & editing:** Isao Goto, Kaori Kikuchi.

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
