## [Decision Letter · Decision Letter 0]

3 Sep 2025

Dear Dr. Goto,

Thank you for submitting your manuscript to PLOS ONE. After careful consideration, we feel that it has merit but does not fully meet PLOS ONE’s publication criteria as it currently stands. Therefore, we invite you to submit a revised version of the manuscript that addresses the points raised during the review process.

We look forward to receiving your revised manuscript.

Kind regards,

Hiroshi Ezura

Academic Editor

PLOS ONE

Journal Requirements:

3. In the online submission form, you indicated that [The data supporting the findings of this study are available upon request from the corresponding author. The data will be provided to interested researchers upon request to ensure transparency and reproducibility of the research, in line with PLOS ONE’s data availability policy.].

5. We notice that your supplementary tables are included in the manuscript file. Please remove them and upload them with the file type 'Supporting Information'. Please ensure that each Supporting Information file has a legend listed in the manuscript after the references list.

Reviewers' comments:

Reviewer's Responses to Questions

**Comments to the Author**

1. Is the manuscript technically sound, and do the data support the conclusions?

Reviewer #1: Partly

2. Has the statistical analysis been performed appropriately and rigorously?

Reviewer #1: Yes

3. Have the authors made all data underlying the findings in their manuscript fully available?

Reviewer #1: Yes

4. Is the manuscript presented in an intelligible fashion and written in standard English?

Reviewer #1: Yes

Reviewer #1: The manuscript “Evaluation of Conditional Treatment Effect of Salt Stress on Tomato Sugar Content

Using Causal Machine Learning: A Pilot Study” (manuscript ID: PONE-D-25-38675) by Goto et al. describes the interaction effects of salinity treatment and environmental factors on tomato fruit sweetness (Brix). The investigation was based on causal inference analysis using causal tree method. Using the method, they divided data into 7 subgroups (data in each subgroup had similar values of variables such as relative humidity) and calculated the impact of salinity treatment on the increase in the probability of fruits with Brix>6. They found that relative humidity lower than 79% is the most influential environmental condition and that the probability was greatly increased by salinity treatment in the condition.

It would help readers of PLOS One to understand the complex mechanism of tomato Brix increase by salinity treatment. However, analysis and discussion are not sufficient. My major requests are as follows.

1. Illuminance and relative humidity are not commonly used for the analysis of environmental responses of plants. Solar radiation [W m−2], PAR [W m−2] and PPFD [µmol m−2 s−1] are commonly used as light strength for the analysis of photosynthesis. I recommend converting illuminance into solar radiation or PAR. Vapor pressure deficit [kPa] and humidity deficit [g m−3] are commonly used as humidity for the analysis of transpiration. I recommend adding causal tree analysis including humidity deficit to covariates instead of air temperature and relative humidity.

I also recommend adding analysis using daytime values as covariates for the causal tree analysis. Because transpiration (and photosynthesis) occurs mainly during the daytime, daytime environmental conditions should be focused on to analyze the environmental effects on stress responses.

2. Because Brix in Control is not stable in this study, not only conditional average of treatment effect (CATE) but also Brix in Control and Treatment should be discussed. Small CATE can include 2 different situations: (1) low Brix in Control and Treatment and (2) high Brix (Brix > 6) in both. Situation (2) occurs under heat conditions and is likely the predominant scenario for subgroups 1 and 2 in this study. A box plot should be added to compare Brix between C/T and among subgroups.

Discussion should be reconsidered. In the 2nd paragraph in Discussion, relative humidity lower than 79%, the cause of high CATE in this study, was associated with the excessive transpiration under lower humidity in the previous studies. However, I think it might be opposite. I think that excessive transpiration occurred during the daytime in subgroups 1 and 2, resulting in the situation (2). Subgroups 1 and 2 consists of mainly cultivation period of C–F, when anthesis occurred from May to August. During the period, transpiration rate often become excessively high due to a high vapor pressure deficit during the daytime even if mist cooling works.

3. Environmental control during the cultivation periods A–I should be described. It can be used for the explanation of the difference in environmental conditions between day/night and among the cultivation periods.

The period when spraying plant hormone (2-methyl-4-chlorophenoxyacetic acid) should be clearly described. And the effect of the plant hormone on fruit Brix should be described and discussed.

I recommend adding information of plant canopy such as LAI for the better understanding of the stress conditions.

Some other issues are given below.

[Abstract]

• The meaning of CATE in this study (the probability of fruits with Brix>6?) should be described in detail.

• “For example, the estimated Conditional Average Treatment Effect (CATE) reached as high as 0.88 when humidity was ≥75%, illuminance was ≥8144 lx, and categorical variable C3 was F1 or F2.” According to Fig. 4, it looks 0.88 resulted from humidity<75%, illuminance<8144 lx, and categorical variable C3 was F1 or F2. There are similar mistakes in Result.

[Materials and methods]

P6

• “ESPSC MIC CO., Ltd.” ESPEC MIC Corp.

• “Temperature and humidity sensors were installed in a ventilation pipe placed inside the canopy” Please attach photos to explain the locations of temperature and humidity sensors on ventilation pipe.

P7

• Table 2 In Fig. 1, n=284 (control) and n=349 (treatment) before propensity score matching, and n=284 (control) and n=284 (treatment) after it. Which is correct, Table 2 or Fig. 1?

• “Truss (C3)” What is the meaning of “C3”?

• The values of Brix and anthesis to harvest are mean (and SD)? If so, please write “mean (SD)” on the rows.

P10 (Propensity score matching)

• “We performed 1-to-1 propensity score matching without replacement to pair fruits in the control and treatment groups while adjusting for the covariates listed in Table 2.” Covariates except for Brix? Please list all variables used for propensity score matching.

[Results]

P18

• “(S1 Table). These findings suggest that environmental factors,” S2 Table?

P21

• “(EC4, 25 mM)” 4.25

[Discussion]

P22

• “Prudent et al. [29] reported co-localized quantitative trait loci with opposing effects on fruit weight and sugar content, suggesting a genetic trade-off under high sink demand.” I think that invoking genetic features at this point is overgeneralization. I recommend removing the sentence.

**Do you want your identity to be public for this peer review?** For information about this choice, including consent withdrawal, please see our Privacy Policy

Reviewer #1: No

---

## [Author Response · Author response to Decision Letter 1]

31 Oct 2025

Referees’ Comments & the Authors’ Responses

Comments from Referee #1:

Comment 1. Illuminance and relative humidity are not commonly used for the analysis of environmental responses of plants. Solar radiation [W m−2], PAR [W m−2] and PPFD [µmol m−2 s−1] are commonly used as light strength for the analysis of photosynthesis. I recommend converting illuminance into solar radiation or PAR. Vapor pressure deficit [kPa] and humidity deficit [g m−3] are commonly used as humidity for the analysis of transpiration. I recommend adding causal tree analysis including humidity deficit to covariates instead of air temperature and relative humidity.

I also recommend adding analysis using daytime values as covariates for the causal tree analysis. Because transpiration (and photosynthesis) occurs mainly during the daytime, daytime environmental conditions should be focused on to analyze the environmental effects on stress responses.

Response 1. Thank you for your valuable suggestion. We agree that using light and humidity metrics commonly adopted for plant environmental response analysis is crucial for enhancing the scientific validity of our study.

We have addressed this comment by implementing the following changes:

1. Light Metric Conversion: We converted Illuminance into PAR (Photosynthetically Active Radiation) for use in the re-analysis.

2. Humidity Metric Conversion and Use: We calculated the Vapor Pressure Deficit (VPD) from relative humidity and air temperature and used it as the primary humidity metric.

3. Daytime Data Focus: To focus on the environmental conditions during the period of active plant physiological response, we used the average values of all environmental covariates during the daytime (6:00 to 18:00).

Based on these changes, we re-ran the Causal Tree Analysis.

【Handling of Results and Explanation】

• In the Main Text, following your recommendation, we have replaced the previous results with the primary analysis using the model that includes VPD and PAR as covariates.

• To further demonstrate the robustness of our findings and compare them with the initial model, we conducted re-analyses under multiple covariate conditions (e.g., models including temperature and relative humidity along with VPD, etc.).

We have applied these revisions consistently throughout the manuscript.

We believe these revisions strengthen our study by grounding our findings in widely accepted plant physiological metrics.

Comment 2. Because °Brix in Control is not stable in this study, not only conditional average of treatment effect (CATE) but also °Brix in Control and Treatment should be discussed. Small CATE can include 2 different situations: (1) low °Brix in Control and Treatment and (2) high °Brix (°Brix > 6) in both. Situation (2) occurs under heat conditions and is likely the predominant scenario for subgroups 1 and 2 in this study. A box plot should be added to compare °Brix between C/T and among subgroups.

Discussion should be reconsidered. In the 2nd paragraph in Discussion, relative humidity lower than 79%, the cause of high CATE in this study, was associated with the excessive transpiration under lower humidity in the previous studies. However, I think it might be opposite. I think that excessive transpiration occurred during the daytime in subgroups 1 and 2, resulting in the situation (2). Subgroups 1 and 2 consists of mainly cultivation period of C–F, when anthesis occurred from May to August. During the period, transpiration rate often become excessively high due to a high vapor pressure deficit during the daytime even if mist cooling works.

Response 2. Thank you for this insightful and critical comment. We agree that examining the stability of °Brix in the Control group, in conjunction with the Treatment group, is essential for a robust interpretation of the Conditional Average Treatment Effect (CATE) estimates.

We have addressed this comment by implementing the following measures:

1. Enhancement of Data Presentation: Addition of a Box Plot

As requested, we have added a Box Plot to the results section (see, page 22, Fig 5), which compares the °Brix values between the Control (C) and Treatment (T) groups across the identified subgroups. This figure clearly demonstrates that the subgroups with small CATE (Subgroups 2, in particular) fall into your proposed Situation (2): high °Brix in both C and T groups. This visualization confirms that a small CATE in these subgroups may reflect a situation where °Brix levels are already high and stable.

2. Reconsideration of the Discussion

This study was approached as a preliminary investigation focused solely on statistically identifying the environmental conditions that appear to modulate the effect of NaCl treatment on tomato fruit sweetness using data analysis and causal tree modeling. We must emphasize that the scope of this work was strictly limited and did not extend to elucidating the detailed underlying physiological mechanisms of action. We fully recognize that understanding the "why"—how these environmental factors influence plants at a mechanistic level—requires dedicated plant physiological studies. Rather, our objective was to provide evidence for a potential causal relationship between the intervention (NaCl) and the outcome (°Brix), conditioned on the environment. We employed the causal tree approach—a machine learning framework designed to automatically identify conditions where the Conditional Average Treatment Effect (CATE) differs—specifically to explore the environmental strata where the treatment effect is most pronounced. While the causal tree statistically partitions the data based on observed covariates, we are mindful that this method cannot directly infer or reveal the underlying physiological processes. Nevertheless, its properties allow us to precisely identify the specific environmental conditions under which stratifying the data will increase the magnitude of the treatment effect.

In the initial version of the manuscript, we discussed why certain conditions (particularly humidity) were selected as important variables, interpreting them in relation to plant transpiration and water uptake. As a result, the reviewer pointed out that “the physiological interpretation should be refined.” We acknowledge that, rather than developing speculative physiological interpretations, it would have been more appropriate to describe the observed relationships more carefully and precisely. At the same time, as the reviewer correctly noted, using covariates that more accurately represent plant physiological responses—such as vapor pressure deficit (VPD)—could provide useful insights into the background mechanisms of the treatment effects.

Therefore, we conducted an additional analysis using a modified set of explanatory variables (Fig. 4; Tables 1, 3, and 4; Supplementary Table 2; and additional Fig. 5). In this reanalysis, the covariates were selected based on the experimental limitations and included the average temperature during the growth period, the daytime average VPD (calculated from daytime temperature and relative humidity), the average photosynthetically active radiation (PAR), and the truss level. The covariates were chosen while considering potential confounders and mediators. Although it is technically possible to increase the number of covariates through feature engineering, unlike in predictive modeling, including all measurable or derived variables can bias the estimated treatment effects, cause overfitting, and reduce reproducibility. Therefore, we limited the analysis to the selected covariates. In addition, following Principle 4 of the American Statistical Association’s (ASA) 2016 Statement on p-values, we predetermined the covariates used in the analysis to avoid arbitrary modification of the analytical plan and prevent p-hacking.

As a result of this reanalysis, a clearer relationship between stratified environmental conditions and treatment effects was obtained, providing more robust insights that could also support further physiological interpretation.

Comment 3. Environmental control during the cultivation periods A–I should be described. It can be used for the explanation of the difference in environmental conditions between day/night and among the cultivation periods.

The period when spraying plant hormone (2-methyl-4-chlorophenoxyacetic acid) should be clearly described. And the effect of the plant hormone on fruit °Brix should be described and discussed.

I recommend adding information of plant canopy such as LAI for the better understanding of the stress conditions.

Response 3. Thank you for your constructive suggestions. We have made the following revisions to address these points:

1. Environmental control: (see, p7, line 5)

We added a description of the automatic environmental control in the Materials and Methods section.

Throughout all cultivation periods, the greenhouse was automatically ventilated when the air temperature exceeded 25 °C, and heating was activated to maintain temperatures above 10 °C in winter. As shown in Table 3, the mean daily air temperature, vapor pressure deficit (VPD), and photosynthetically active radiation (PAR) during the fruiting period differed clearly among cultivation periods, reflecting seasonal environmental variations.

2. Plant hormone application: (see page 6, line 11).

We have added a detailed description of the 2-methyl-4-chlorophenoxyacetic acid (4-CPA) treatment in the Materials and Methods section.

Once the first flower on each truss had fully opened, all flowers on the truss were sprayed with a commercial 4-CPA formulation (Tomato-Ton; Green Japan Co., Ltd., Japan). The solution was diluted to concentrations of approximately 30 mg L⁻¹ at low temperatures (<20 °C) and 15 mg L⁻¹ at high temperatures (≥20 °C), following the manufacturer’s instructions.

Because the same procedure and concentration adjustments were uniformly applied to both control and NaCl treatment groups, the 4-CPA application was not expected to differentially affect fruit °Brix.

3. Plant canopy information (LAI): (see page 6, line 17).

We added information about the canopy structure in the Materials and Methods section.

The plants were spaced at 30 cm within rows, resulting in a planting density of 2,380 plants per 10 a. The cultivation was conducted under sparse planting conditions, and the leaf area index (LAI) during the fruit development period was approximately 3.5. Because plant management (pruning, training, and leaf retention) was consistent across treatments, differences in canopy density and light interception were assumed to be minimal.

Thank you for pointing out these minor errors. We have corrected them as requested below.

[Abstract]

Minor Comment 1: “The meaning of CATE in this study (the probability of fruits with °Brix>6?) should be described in detail.”

Response to Reviewer #1, Minor Comment 1:

Thank you for this constructive comment. We agree that a clearer definition of CATE (Conditional Average Treatment Effect) is essential for the readability of the paper. We have now explicitly defined CATE, which in this study represents the proportion of fruits with °Brix > 6%, and ensured this definition is thoroughly explained in the Methods and Estimation of CATE section (see page　2, line 14 and page　15, line 13).

Minor Comment 2: “For example, the estimated Conditional Average Treatment Effect (CATE) reached as high as 0.88 when humidity was ≥75%, illuminance was ≥8144 lx, and categorical variable C3 was F1 or F2.” According to Fig. 4, it looks 0.88 resulted from humidity<75%, illuminance<8144 lx, and categorical variable C3 was F1 or F2. There are similar mistakes in Result.

Response to Reviewer #1, Minor Comment 2:

We appreciate the reviewer's careful reading and identifying the factual inconsistency concerning the CATE conditions. We agree this was a significant mistake and apologize for the error.

It is important to note that our response to Major Comment 2 required us to change the covariates and conduct a complete re-analysis. Due to this fundamental change in the methodology and results, the description of the CATE condition has been naturally superseded.

Therefore, we have removed and fully rewritten all relevant sections of the manuscript, including the Results section, to ensure full consistency with the new analysis (see page 2, line 16 and page 21, line 6.

[Materials and methods]

Minor Comment 3: P6 “ESPSC MIC CO., Ltd.” ESPEC MIC Corp.

Response to Reviewer #1, Minor Comment 3:

Thank you for pointing out the typo in the company name. We appreciate your attention to detail. We have corrected "ESPSC MIC CO., Ltd." to the proper name, "ESPEC MIC Corp." The correction can be found on page 7, line 10.

Minor Comment 4: “Temperature and humidity sensors were installed in a ventilation pipe placed inside the canopy” Please attach photos to explain the locations of temperature and humidity sensors on ventilation pipe.

Response to Reviewer #1, Minor Comment 4:

We thank the reviewer for this excellent and constructive suggestion. We agree that visual clarification of the sensor setup is essential for the reader to fully understand the measurement methodology.

As requested, we have attached a separate reference photograph that clearly shows the installation of the temperature and humidity sensors inside the ventilated PVC tubes within the plant canopy (see SensorInstallation_Example2.jpg).

SensorInstallation_Example2.jpg

Furthermore, the reviewer's comment correctly highlighted that the original text lacked sufficient detail regarding the technical design of the ventilated pipe (tube) and the sensor placement mechanism. We have therefore thoroughly revised the corresponding sentence in the Methods section to provide necessary context, technical specifications, and a supporting reference.

The revised text now reads:

Temperature and humidity sensors were installed inside a ventilated PVC tube (φ5 cm, double-tube type) equipped with a small DC fan to ensure forced airflow, following the design principle described by Okada and Nakamura (2010). This setup was placed horizontally within the plant canopy to measure air temperature and humidity under well-ventilated conditions while minimizing the influence of solar radiation (see page 7, line 9).

We have also clarified that the data from the sensor closest to the analyzed plant were used for the temperature, humidity, and light intensity measurements.

Temperature and humidity sensors were installed at multiple locations within the plant canopy, and data from the sensor positioned closest to the experimental plants were used for analysis.

Minor Comment 5: P7 Table 2 In Fig. 1, n=284 (control) and n=349 (treatment) before propensity score matching, and n=284 (control) and n=284 (treatment) after it. Which is correct, Table 2 or Fig. 1?

Response to Reviewer #1, Minor Comment 5:

Thank you for pointing out this inconsistency in the sample sizes (n) between Table 2 and Figure 1. We apologize for this oversight.

Due to the comprehensive re-analysis that we performed, all sample sizes and numerical values have been updated and are now consistent throughout the manuscript (see page 9, Table 2 and Figure 1).

Minor Comment 6: “Truss (C3)” What is the meaning of “C3”?

Response to Reviewer #1, Minor Comment 6:

Thank you for raising this question regarding the term "C3."

We acknowledge that the inclusion of this term caused confusion. "C3" was an internal, automatically assigned variable code used solely during the data analysis process and holds no specific scientific meaning in the context of the manuscript.

To eliminate any ambiguity and maintain clarity, we have completely removed the superfluous designation "C3" and ensured that this variable is consistently and solely referred to as "Truss" throughout the entire manuscript.

This correction is reflected on page 10 and in Table 2.

Minor Comment 7: The values of °Brix and anthesis to harvest are mean (and SD)? If so, please write “mean (SD)” on the rows.

Response to Reviewer #1, Minor Comment 7:

Thank you for this helpful comment regarding the data representation in the table. We confirm that the values for °Brix and anthesis to harvest are indeed presented as Mean (

---

## [Decision Letter · Decision Letter 1]

19 Nov 2025

Dear Dr. Goto,

Thank you for submitting your manuscript to PLOS ONE. After careful consideration, we feel that it has merit but does not fully meet PLOS ONE’s publication criteria as it currently stands. Therefore, we invite you to submit a revised version of the manuscript that addresses the points raised during the review process.

We look forward to receiving your revised manuscript.

Kind regards,

Hiroshi Ezura

Academic Editor

PLOS ONE

Journal Requirements:

Reviewers' comments:

Reviewer's Responses to Questions

**Comments to the Author**

Reviewer #1: All comments have been addressed

2. Is the manuscript technically sound, and do the data support the conclusions?

Reviewer #1: Yes

3. Has the statistical analysis been performed appropriately and rigorously?

Reviewer #1: Yes

4. Have the authors made all data underlying the findings in their manuscript fully available?

Reviewer #1: Yes

5. Is the manuscript presented in an intelligible fashion and written in standard English?

Reviewer #1: Yes

Reviewer #1: Comment 1. Brief explanation of Fig. 5 should be described in Result.

Comment 2. Conclusion “Causal Tree analysis revealed that temperature and VPD jointly affected the treatment effect, with the strongest enhancement occurring under moderate conditions (20–25 °C, VPD < 0.84 kPa).” The condition 20–25 °C and VPD < 0.84 kPa caused minimum enhancement 0.031. For the strongest enhancement, the conditions was temperature < 20, VPD >= 0.84, and PAR < 13. Please revise the sentence. I apologize if I have misunderstood.

**Do you want your identity to be public for this peer review?** For information about this choice, including consent withdrawal, please see our Privacy Policy

Reviewer #1: No

---

## [Author Response · Author response to Decision Letter 2]

15 Dec 2025

Comments from Referee #1:

Comment 1. Brief explanation of Fig. 5 should be described in Result.

Response 1. Thank you for pointing this out. We agree with the reviewer’s suggestion. We have added a brief explanation of Fig. 5 in the Results section (page 21, line 10) to ensure that the figure is properly introduced and described.

Comment 2. Conclusion “Causal Tree analysis revealed that temperature and VPD jointly affected the treatment effect, with the strongest enhancement occurring under moderate conditions (20–25 °C, VPD < 0.84 kPa).” The condition 20–25 °C and VPD < 0.84 kPa caused minimum enhancement 0.031. For the strongest enhancement, the conditions was temperature < 20, VPD >= 0.84, and PAR < 13. Please revise the sentence. I apologize if I have misunderstood.

Response 2. Thank you very much for this insightful comment. After reviewing the figure again, we confirmed that the reviewer’s interpretation was correct and that our original description was inaccurate. We have therefore revised the relevant sentences in the Conclusion section (page 28, line 10-13). to accurately describe the conditions associated with the strongest enhancement. We appreciate the reviewer’s careful examination of our work..

---

## [Editor Report · Decision Letter 2]

21 Dec 2025

Evaluation of Conditional Treatment Effect of Salt Stress on Tomato Sugar Content Using Causal Machine Learning: A Pilot Study

PONE-D-25-38675R2

Dear Dr. Goto,

We’re pleased to inform you that your manuscript has been judged scientifically suitable for publication and will be formally accepted for publication once it meets all outstanding technical requirements.

Kind regards,

Hiroshi Ezura

Academic Editor

PLOS One
---

## [Editor Report · Acceptance letter]

PONE-D-25-38675R2

PLOS One

Dear Dr. Goto,

I'm pleased to inform you that your manuscript has been deemed suitable for publication in PLOS One. Congratulations! Your manuscript is now being handed over to our production team.

Kind regards,

on behalf of

Prof. Hiroshi Ezura

Academic Editor

PLOS One